# Characterisation of protein isoforms encoded by the *Drosophila* Glycogen Synthase Kinase 3 gene *shaggy*

Dagmara Korona[1], Daniel Nightingale[2], Bertrand Fabre[2], Michael Nelson[3], Bettina Fischer[1], Glynnis Johnson[1], Jonathan Lees[4], Simon Hubbard[3], Kathryn Lilley[2], Steven Russell[1]*

**1** Department of Genetics, University of Cambridge, Cambridge, United Kingdom, **2** Department of Biochemistry, Cambridge Centre for Proteomics, University of Cambridge, Cambridge, United Kingdom, **3** Faculty of Biology, Medicine and Health, Manchester Academic Health Science Centre Manchester, University of Manchester, Manchester, United Kingdom, **4** Department of Biological and Medical Sciences, Faculty of Health and Life Sciences, Oxford Brookes University, Oxford, United Kingdom

* s.russell@gen.cam.ac.uk

**Data Availability Statement:** Relevant data are within the manuscript and its Supporting Information files. RNAseq data https://www.ncbi.nlm.nih.gov/geo/query/acc.cgi?acc=GSE139040.

## Abstract

The *Drosophila shaggy* gene (*sgg*, *GSK-3*) encodes multiple protein isoforms with serine/threonine kinase activity and is a key player in diverse developmental signalling pathways. Currently it is unclear whether different Sgg proteoforms are similarly involved in signalling or if different proteoforms have distinct functions. We used CRISPR/Cas9 genome engineering to tag eight different Sgg proteoform classes and determined their localization during embryonic development. We performed proteomic analysis of the two major proteoform classes and generated mutant lines for both of these for transcriptomic and phenotypic analysis. We uncovered distinct tissue-specific localization patterns for all of the tagged proteoforms we examined, most of which have not previously been characterised directly at the protein level, including one proteoform initiating with a non-standard codon. Collectively, this suggests complex developmentally regulated splicing of the *sgg* primary transcript. Further, affinity purification followed by mass spectrometric analyses indicate a different repertoire of interacting proteins for the two major proteoforms we examined, one with ubiquitous expression (Sgg-PB) and one with nervous system specific expression (Sgg-PA). Specific mutation of these proteoforms shows that Sgg-PB performs the well characterised maternal and zygotic segmentations functions of the *sgg* locus, while Sgg-PA mutants show adult lifespan and locomotor defects consistent with its nervous system localisation. Our findings provide new insights into the role of GSK-3 proteoforms and intriguing links with the GSK-3α and GSK-3β proteins encoded by independent vertebrate genes. Our analysis suggests that different proteoforms generated by alternative splicing are likely to perform distinct functions.

**Funding:** BBSRC grant (BB/L002817/1) to SH, KL and SR. The funders had no role in study design, data collection and analysis, decision to publish, or preparation of the manuscript.

**Competing interests:** The authors have declared that no competing interests exist.

## Introduction

Glycogen Synthase Kinase-3 (GSK-3) is a highly conserved protein kinase that has orthologs in all metazoans, with proteins from distant species such as flies and humans displaying more than 90% sequence similarity in the protein kinase domain [1, 2]. Initially identified as an enzyme involved in the regulation of glycogen metabolism, a key role for the *Drosophila* orthologue encoded by the *shaggy* (*sgg*) locus in embryonic segmentation [3] established GSK-3 at the heart of the Wnt signalling pathway in flies and vertebrates [4]. In brief, GSK-3 kinase activity acts to negatively regulate Wnt signalling by phosphorylating β-catenin, Armadillo (Arm) in *Drosophila*, such that it is ubiquitinylated and subsequently degraded by the proteasome. When Wnt signalling is active, GSK-3 is inactivated, Arm is stabilized and translocates to the nucleus where it binds to the Tcf transcription factor to activate Wnt responsive genes [5]. Considerable work from many laboratories has established that GSK-3 and Wnt signalling is pivotal for cell differentiation and morphogenesis across the Metazoa [5, 6].

In vertebrates, there are two major isoforms of GSK-3, alpha and beta, each encoded by independent paralogous genes. While these isoforms share considerable homology in the kinase domain (85% overall identity, 98% within the kinase domains) [7], they show major differences at their termini with GSK-3α containing a large glycine-rich N-terminal region that is absent in GSK-3β [2]. Loss of GSK-3β in mice results in late embryonic lethality with liver, cardiac and craniofacial defects [8–10], and in Xenopus, expression of a kinase dead version of GSK-3β resulted in axis formation defects [11]. In addition, GSK-3β heterozygous mice exhibit a range of phenotypes, particularly in aspects of metabolism, homeostasis and nervous system function [12–15]. In contrast, loss of function GSK-3α mice are viable but show alterations in glucose metabolism [16] and abnormalities in brain structure and behaviour [17, 18]. Interestingly, there is evidence that the mammalian isoforms show both partially redundant and antagonistic interactions [12, 19–21] with the current view that they act at least partially redundantly in early embryonic Wnt signalling [22]. In contrast to vertebrates, the *Drosophila* genome contains a single major GSK-3 locus, *sgg*, that shows considerable complexity, with 17 annotated transcripts encoding 10 different protein isoforms predicted (Fig 1A and 1B). A second GSK-3 enzyme is encoded by the *gasket* locus, a retrotransposed gene whose expression is largely restricted to the testis [23] and is not considered further here. Sgg proteoforms differ at their N termini (5 alternatives), at internal exons and at the C terminus (3 alternatives). At the C terminus, Sgg-PD is unique among the proteoforms and was previously identified as Sgg46 [24]. The remaining nine proteoforms containing either a short C terminus, typified by Sgg-PB (Sgg10), or a longer C terminus typified by Sgg-PA (Sgg39). The longer isoform contains a glycine-rich region that is analogous to the N-terminal domain of the vertebrate GSK-3α. The role of this domain is currently not well understood but is predicted to contain an ANCHOR binding region [25] and two short MATH domain interaction motifs, both thought to be important in protein interactions [26] (Fig 1C).

In *Drosophila*, *sgg* is known to have a variety of developmental roles and interacts with a number of signalling pathways including Wnt, Hedgehog, Notch and Insulin [27–29], as well as being implicated in a variety of other cellular processes [30]. Null mutations in *sgg* exhibit a maternal effect lethal phenotype with a strong segment polarity defect in embryos lacking zygotic and maternal Sgg, as well as defects in the central and peripheral nervous systems [31]. In addition, analysis of a wide range of other alleles has reveal phenotypes in diverse tissues, for example in the macrochaetes, mechanosensory bristles found on the adult thorax, where it has been shown to phosphorylate key transcription factors [32]. However, despite the considerable focus on the role of *sgg* in development, little is known about how particular proteoforms contribute to specific functions. Previous work indicates that Sgg-PB (Sgg10) is an

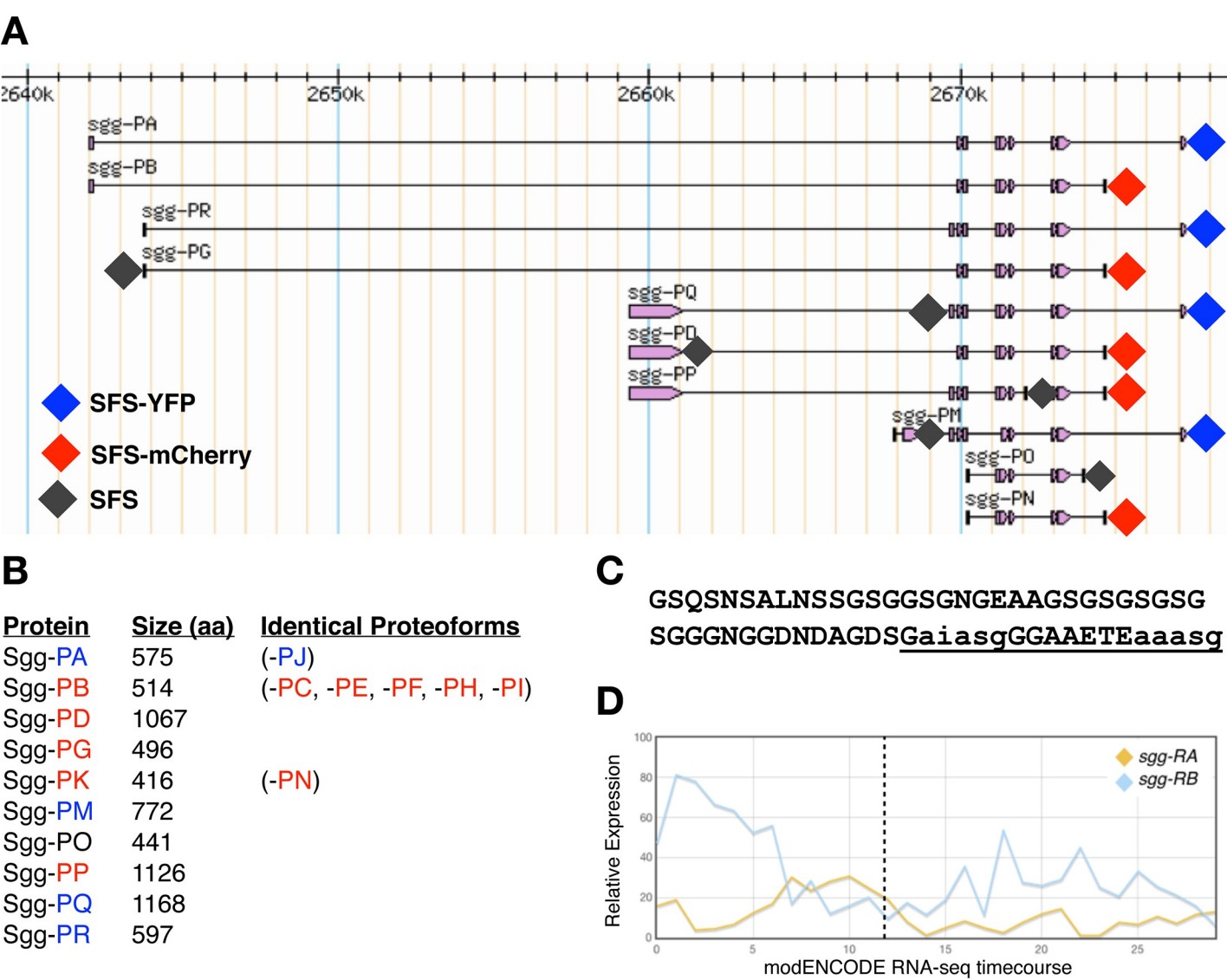

**Fig 1. A)** simplified map of *shaggy* locus showing the major proteoforms (from FlyBase). Diamonds indicate the position of insertion sites tagged by CRISPR/Cas9 with the colour representing the tag used. **B)** Length of each of the major Sgg proteoforms in amino acids and an indication of proteoforms sharing the same amino acid sequence (from FlyBase). **C)** Amino acid sequence of the C terminal exon differentiating isoform A from isoform B. The underlined sequence is a predicted ANCHOR binding region, lowercase letters indicate predicted MATH domain interaction motifs. **D)** modENCODE RNAseq timecourse of relative expression levels of transcripts encoding the major Sgg-PA Sgg-PB isoforms. The X axis indicates the modENCODE samples collected with the first 12 representing 2hr intervals across embryogenesis. Dotted line indicates the end of embryonic development.

important proteoform, maternally contributed and detected throughout development into adults [33]. In contrast, Sgg-PA (Sgg39) has more limited expression, it does not appear to be maternally contributed (Fig 1D). and is not detected in wing imaginal disks, where the adult macrochaetes develop. A third major isoform, Sgg-PD (Sgg46), contains a C-terminal domain that includes a caspase-cleavage site. It appears to be dispensable for viability but has a role in sensory organ precursor development [24, 33]. Rescue experiments in *Drosophila* indicate that expression of a Sgg-PB isoform can fully rescue *sgg* null phenotypes and that mammalian GSK-3β but not GSK-3α can provide partial rescue of some, but not all null phenotypes,

emphasising the difference between the mammalian genes and pointing to functional differences between Sgg proteoforms [28].

The complexity of the *sgg*, locus in *Drosophila* with its multiple proteoforms and the apparent differences between GSK-3 paralogs in vertebrates raises the question of how different GSK-3 proteoforms contribute to the functions of this key kinase during development and in homeostasis. In particular, there has been a general debate as to whether protein isoforms encoded by the multiple splice forms of a particular gene are produced and functional. One extreme, based on evidence from high throughput mass-spectrometry or literature curation of verified proteoforms, contends that most genes encoding multiple alternatively spliced isoforms only produce a single functional proteoform [34, 35]. In contrast, an alternative view is that alternatively spliced isoforms generate proteoforms with functionally distinct properties in terms of spatial or temporal expression, or their interaction repertoires [36, 37]. To help address the role of alternative Sgg proteoforms in *Drosophila* and the developmental roles GSK-3 plays, as well as contributing to the debate surrounding the functionality of splice isoforms, we used a CRISPR-Cas9 based genome engineering strategy to tag specific Sgg proteoforms. We introduced fluorescent protein or affinity tags into the endogenous *sgg* locus in *Drosophila* [38], altogether tagging eight different exons. This allowed us to follow the expression of Sgg proteoforms across embryonic development by immunohistochemistry and/or fluorescence microscopy, revealing unique and specific expression for each of the tagged proteoforms. Focusing on the major C terminal domains, we show that the short form (Sgg-PB) is ubiquitously expressed across embryogenesis and is essential for viability. In contrast, the long form (Sgg-PA) is specifically expressed in the developing nervous system and is not required for viability. Furthermore, using the tagged lines to identify interacting proteins for each proteoform class we found a different set of interactors. This agrees with an analysis of mammalian GSK-3α and β interactions using a yeast 2-hybrid approach which found a different set of interacting proteins for these closely related proteins [39]. We found that the loss of major proteoforms is not always compensated by other isoforms and can lead to age-related pathologies including accelerated senescence. Taken together, our work suggests that the transcript complexity of the *Drosophila sgg* locus reflects functionally relevant differences in the spatial and temporal expression of GSK-3 as well as functional differences between major proteoforms.

## Results and discussion

### In vivo tagging of major Sgg proteoforms

In order to determine the expression and localisation of specific Sgg proteoforms we used CRISPR/Cas9 genome engineering to introduce different in-frame protein tags into specific exons at the endogenous *sgg* locus [38]. We first focused on the major C terminal proteoforms represented by Sgg-PA and Sgg-PB, constructing fly lines containing a 3xFLAG-StrepTagII-mVenus-StrepTagII (FSVS) cassette just before the termination codon. We have previously utilised this cassette in a large-scale protein trap screen [40] and found it was tolerated by a wide range of different *Drosophila* proteins *in vivo*. In both cases the lines we generated were homozygous or hemizygous viable and fertile. Using an antibody against the FLAG epitope we first examined the expression of each tagged proteoform in the *Drosophila* embryo via immunohistochemistry. With Sgg-PA we found little or no expression during early development but by stage 9 (germband extension) we observed strong and specific expression in the developing CNS of the trunk and then brain. As development proceeded expression became prominent in the elaborating PNS and was particularly strong in the chordotonal organs, where it continued to the end of embryogenesis (Fig 2E and 2H). Looking more closely we observed that in the chordotonal organs staining was associated with the cell bodies and extended into the ciliated

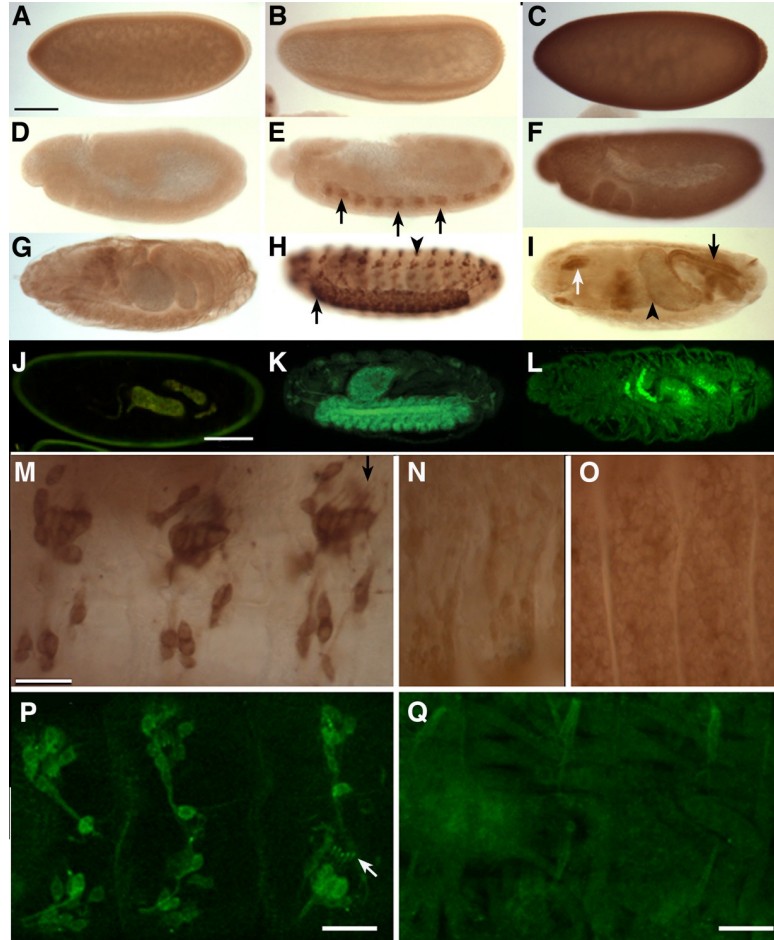

**Fig 2. Immunohistochemistry and live imaging of tagged Sgg-PA and -PB proteoforms.** All stainings are with anti-FLAG. **(A, D and G)** wild type embryos at stage 5–6 (blastoderm), 10–11 (extended germ band) and 16 (late embryogenesis) respectively. **(B, E and H)** tagged Sgg-PA at similar stages showing expression in the developing CNS (arrows in E), mature CNS and PNS (arrow and arrowhead in H). **(C, F and I)** tagged Sgg-PB at equivalent stages showing ubiquitous expression at the blastoderm **(C)** and germband extension stages **(F),** followed by localised expression in hindgut (arrow in I), midgut (arrowhead in I) and salivary glands (white arrow in I). **(J-L)** live confocal images of late stage embryos. Wild type showing gut autofluorescence **(J)**, Sgg-PA^FSVS showing prominent CNS expression **(K)**, Sgg-PB^FSVS showing mesoderm expression **(L)**. **(M)** lateral view of abdominal chordotonal organs from a stage 15 Sgg-PA embryo. **(N and O)** lateral view of the mesoderm **(N)** and epidermis **(O)** from a stage 14 Sgg-PB embryo. **P)** lateral view of YFP expression in abdominal chordotonal organs of a stage 15 Sgg-PA embryo. **(Q)** lateral view of YFP expression the epidermis of a stage 15 Sgg-PB embryo. All embryos are lateral views with anterior to the left. Scale bar in A = 100μm (applies to A-L), Scale bar in M = 20μm (applies to M-Q).

endings. (arrows in Fig 2M and 2P). Towards the end of embryogenesis, we observed specific staining in a subset of cells in the developing brain and in the anterior commissural bundle (S1 Fig). In contrast, we found strong and ubiquitous staining in the Sgg-PB lines in the early embryo, as expected from the strong maternal contribution detected by RNA-seq analysis (Fig 1D), that continued until germ band retraction (Fig 2C and 2F). Demonstrating that the ubiquitous strong staining reflects Sgg-PA expression, we live imaged embryos carrying an mCherry tagged Sgg-PA and observed similar ubiquitous expression in the blastoderm (S1F Fig). At later stages expression was prominent in the hindgut, midgut and salivary glands (Fig 2I). At higher magnification in stage 11–13 embryos (germ band retraction), we observed staining in the developing muscles (Fig 2N) and epidermis (Fig 2O).

**Sgg-PA<sup>YFP</sup>, Sgg-PB<sup>mCh</sup>**  **Sgg-PA<sup>mCh</sup>, Sgg-PB<sup>YFP</sup>**

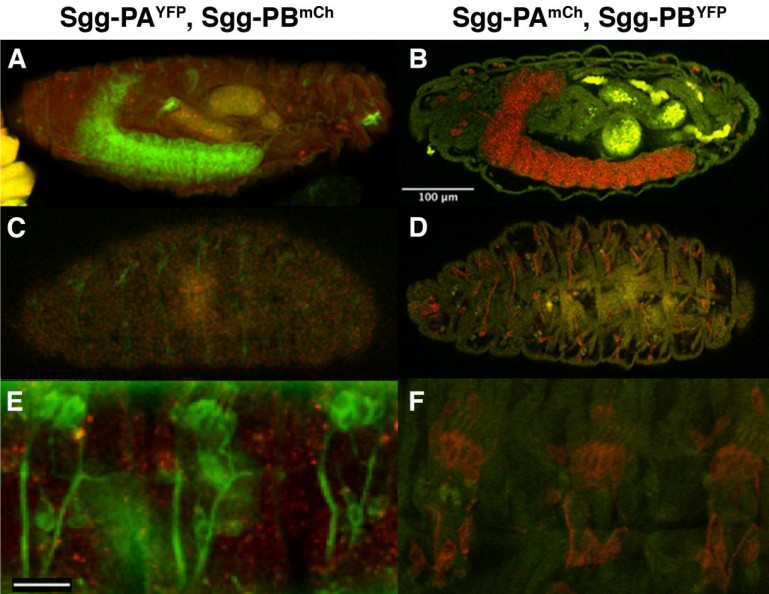

**Fig 3. Fluorescence imaging of tagged Sgg proteoforms. (A and C)** heterozygous stage 17 female embryos carrying Sgg-PA tagged with YFP and Sgg-PB tagged with mCherry showing Sgg-PA expression in CNS **(A)** and PNS **(C)** with weaker ubiquitous expression of Sgg-PB. **(B and D)** stage 17 heterozygous female embryos carrying reciprocally tagged lines (Sgg-PA-mCherry, Sgg-PB-YFP) highlighting Sgg-PA in the CNS **(B)** and PNS **(D)** and Sgg-PB in the mesoderm **(D)**. **(E and F)** close up view of Sgg-PA in chordotonal organs (YFP in E and mCherry in F) and punctate epidermal Sgg-PB (mCherry in E) and mesodermal Sgg-PB (YFP in F). All embryos oriented anterior to the left, dorsal to the top. Scalebar in B = 100μm (applies to A-D), Scale bar in E = 20μm (applies to E and F).

To confirm the localisation observed with the immunohistochemistry, we examined the FSVS tagged versions of Sgg-PA and Sgg-PB by confocal imaging unfixed samples. We found very similar, if not identical, localisation to that obtained by immunohistochemistry (Fig 2K and 2L). In particular the fluorescence allowed much clearer visualisation of Sgg-PA throughout the chordotonal organs (Fig 2P) and Sgg-PB in the musculature (Fig 2Q). We also generated alternatively tagged versions of each proteoform by replacing the YFP tag with mCherry. We generated embryos heterozygous for either Sgg-PA<sup>YFP</sup>/Sgg-PB<sup>mCh</sup> (Fig 3A and 3C) or Sgg-PA<sup>mCh</sup>/Sgg-PB<sup>YFP</sup> (Fig 3B and 3D) and imaged these by confocal microscopy. While in general the mCherry signal was significantly weaker that the YFP, we were able to confirm and extend our immunohistochemistry observations. The fluorescent reporters confirmed the strong localisation of Sgg-PA to the central nervous system and prominently in the PNS as well as Sgg-PB in the epidermis and mesoderm (Fig 3E and 3F), particularly in the epidermis, Sgg-PB appeared to give a more punctate appearance (Figs 3E and S1G).

## Tagging other Sgg proteoforms

We extended our analysis to examine other Sgg proteoforms, introducing exon specific 3xFLAG-StrepTagII tags into the endogenous *sgg* locus (Fig 1, S1 Table). We tagged the C-terminus of the first exon of Sgg-PD (which also tags -PP and -PQ); the first coding exon of Sgg-PG (also tags -PR), the unique terminal exon of Sgg-PO, a unique internal exon of Sgg-PP and an internal exon of Sgg-PQ (shared with -PM, -PP and -PR). Finally, Sgg-PM is predicted to initiate with a valine rather than a methionine and we tagged a unique exon in this proteoform to confirm the translation from a non-standard initiation codon. All of the tagged lines we generated were homozygous or hemizygous viable and fertile, and we again examined

expression in fixed samples by immunohistochemistry with a monoclonal antibody recognising the FLAG epitope (S2 Fig). In contrast to the -PA and -PB tagged lines we noticed that, apart from Sgg-PD, the expression levels of the other proteoforms was generally weaker and mostly restricted to particular tissues.

**Sgg-PD.** This variant shows early expression that is largely restricted to mesoderm (S2D Fig) and becomes more prominent at stage 9 (S2E Fig). At later stages the expression of Sgg-PD is generally weak and ubiquitous with elevated levels found in the CNS, posterior spiracles, Malpighian tubules, proventriculus and salivary glands (S2F and S2V Fig).

**Sgg-PG and -PR.** These proteoforms share a unique amino terminus and have either the long (-PR) or short (-PG) C termini described above, they also differ in a short internal exon but our tagging does not differentiate between these (S1 Table). We did not detect any strong expression during early stages, but by mid-embryogenesis we observed transient mesoderm expression (S2H Fig) followed by later expression in the hindgut (S2I Fig), foregut and the anterior region of the pharynx (S2W Fig).

**Sgg-PM.** This proteoform is predicted to initiate with an unconventional start codon, a valine rather than a methionine, and shares a C terminus with Sgg-PA. We found no early expression of this proteoform. Late in development we observed expression in the pharynx, proventriculus and weakly in the hindgut (S2L Fig).

**Sgg-PO.** This Sgg-PO proteoform has a unique C-terminus (Fig 1), and is not detected during early stages but again shows detectable expression in the mesoderm (S2N Fig) from stage 9 and at later stages in the hindgut, posterior spiracles and prominently in the proventriculus (S2O and S2X Fig).

**Sgg-PP.** This proteoform shares the N-terminal exon of Sgg-PD and the short C-terminal exon but has a short unique exon that we tagged. We did not find any expression during early embryogenesis, although there appears to be faint staining in the mesoderm at stage 9 (S2Q Fig), but we did observe expression in the hindgut and anterior midgut of late embryos (S2R and S2Y Fig).

**Sgg-PQ.** We tagged this proteoform at an internal exon shared with–PR, -PP, and–PM. These proteoforms were not detected during early embryogenesis but at later stages it was is prominent in the salivary glands and proventriculus (S2U Fig).

Taken together, our tagging strategy has revealed dynamic and tissue-specific expression of different Sgg proteoforms during embryogenesis. Our most striking finding is the clear difference in expression of the major C-terminus proteoforms exemplified by Sgg-PA and Sgg-PB, which, as we describe above, may correspond to the GSK-3α and GSK-3β proteins encoded by separate genes in vertebrate genomes. We therefore elected to generate specific loss of function mutations in each of these major isoforms by separately deleting their specific C-terminal exons. A further noticeable feature of the proteoform expression was the localisation of different tagged proteoforms in the developing digestive system, particularly in the proventriculus and hindgut. While it has been shown that *sgg* expression is enriched in particular regions of the adult gut, particularly the crop and hindgut [41], there have been relatively few reports of functional roles for *sgg* in the gut [42, 43] or other specific tissues in the embryo [44]. Finally, the extensive early mesoderm expression shown by several proteoforms is consistent with the previously established role for *sgg* and Wnt signalling in embryonic muscle cell progenitors [45].

## Proteoform specific mutations and transcriptomics

We used CRISPR-Cas9 genome engineering to generate in locus deletions to remove the C-terminal exons specific for the Sgg-PA or Sgg-PB proteoform families. We completely

removed the unique exons, replacing them with a 3Px3 driven DsRED marker, flanked by LoxP sites, that was subsequently removed by the activity of Cre recombinase to leave the remainder of the locus largely unaltered [46]. In this way we generated Sgg-PB mutations, where the following upstream exon unique for isoform Sgg-PA was left intact. Similarly, we removed the last unique exon for isoform Sgg-PA ensuring the flanking sequences were not affected. Both of the exon deletions were confirmed by genotyping via PCR and sequencing.

Deletion of the Sgg-PA class proteoforms ($sgg^{isoA}$) resulted in flies that were viable and fertile, whereas loss of the Sgg-PB class proteoforms ($sgg^{isoB}$) resulted in late embryo/early larval lethality. We note that in the case of $sgg^{isoA}$, progeny from homozygous mothers lack both maternal and zygotic contributions and are thus completely null, whereas progeny from hemizygous $sgg^{isoB}$ mothers have a maternal contribution of wild type transcript or protein.

We examined mutant lines by immunostaining to determine any effects on CNS and PNS development and only observed minor defects in a small percentage of progeny (<5%). Examination of larval cuticles revealed a similar low frequency of defects, with the very occasional appearance of animals resembling *sgg* loss of function phenotypes (<1%). We conclude that the unique C-terminal extension defining the Sgg-PA class of proteoforms is dispensable for normal development, a similar situation as seen in vertebrates with the loss of GSK-3α. We presume that in this case, ubiquitous expression of the shorter Sgg-PB class of proteoforms is able to provide sufficient Sgg function in the nervous system. In contrast, loss of the Sgg-PB class terminal exon is lethal and either cannot be rescued by the longer C-terminus or selection of the Sgg-PA terminal exon is tissue specific and unable to be spliced in some tissues.

To examine whether the loss or reduction of Sgg proteoforms had consequences for gene expression, we performed RNA-seq analysis using RNA extracted from the isoform specific null mutants (Fig 4A, S2 Table). In the case of $sgg^{isoA}$, embryos from homozygous mothers crossed to hemizygous fathers are completely null for the proteoforms containing this exon and we compared RNA from these embryos with stage matched embryos from the progenitor stock. We performed triplicate biological replicates and after filtering (1.6-fold expression change, p<0.05) we identified 100 genes with significantly changed expression (26 up and 74 down) with no significant enrichment of any Gene Ontology terms apart from a down-regulation of 6 mitochondria-encoded respiratory chain components (*ATPase-6*, *Cyt-b* and 4 ND subunits) along with 7 other enzymes involved in respiration or redox reactions. Given that $sgg^{isoA}$ null embryos are viable and fertile the lack of any major effects on gene expression was not unexpected.

For the analysis of $sgg^{isoB}$ we extracted RNA from null hemizygous male embryos derived from heterozygous mothers, identifying these individuals by the lack of a GFP balancer, and compared this with RNA from the progenitor stock. While we expect a rescue of the zygotic mutation by the maternal component, we nevertheless identified 482 genes with significant expression changes (94 up and 388 down) (S2 Table). In particular, we noted a strong upregulation of Heat Shock Factor (Hsf) and a number of stress response and chaperone genes, but down regulation of sets of genes implicated in cuticle development and proteolysis. Many of these dysregulated genes form a highly connected network (p <10e-16, Fig 4B) indicating that mutants are clearly perturbed at the transcriptional level. We presume these gene expression changes reflect the gradual loss of $sgg^{isoB}$ maternal product, however, and in line with the lack of overt morphological phenotypes, we note that we found no apparent changes in any major developmental or signalling pathways.

## Proteoform-specific interactions

Since the two major Sgg proteoform classes are expressed in spatially different patterns it is possible that they participate in different pathways, have different protein partners or

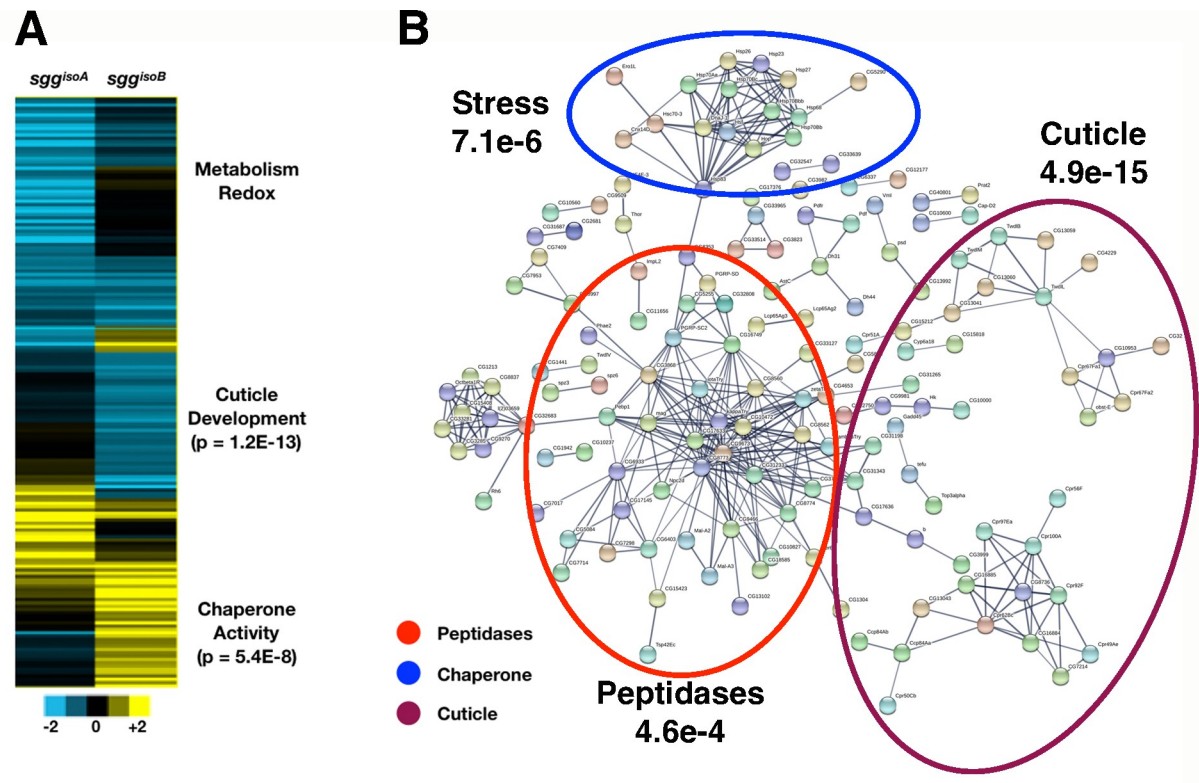

**Fig 4. RNA-seq of *sgg* isoform mutants. (A)** heatmaps of significant expression changes in embryos maternally and zygotically null for *sgg*[isoA] and zygotically null for *sgg*[isoB] with significant GO enrichments indicated. (**B**) String interaction map constructed from genes with significantly changed expression in *sgg*[isoB] zygotic nulls showing highly connected networks of peptidases, chaperone functions and cuticle biosynthesis genes (p-values indicated corrected gene ontology enrichments for the indicated terms).

preferentially act on a different spectrum of substrates. To gain insight into possible unique roles we performed immunopurifications followed by mass spectrometry analysis to identify Sgg-PA and Sgg-PB interactomes. Using our previously described IPAC approach [47], we performed independent purifications using the StrepII, FLAG and YFP tags introduced into the *sgg* locus to increase the reliability of the identified interacting partners. In parallel, we also used a protein trap line we had previously generated (*sgg*[CPTI002603]) that appears to tag the majority of Sgg proteoforms [40], along with a *w*[1118] negative control. We applied the QProt pipeline, a tool for examining differential protein expression, with a p-value cut-off of 0.05 and a requirement that putative interactors be identified in at least 2 out of the 3 independent pull downs after correcting against the wild type control. We identified 20 co-purifying proteins with Sgg[CPTI002603], 26 with the Sgg-PA and 21 with Sgg-PB (Table 1).

We first determined whether the lists of potential interactors were enriched for any Gene Ontology terms and found that the Sgg-PA and Sgg[CPTI002603] lists were significantly enriched for processes involved in ribosome assembly, cytoplasmic translation and protein folding (adjusted p <2e-06), whereas the Sgg-PB list showed no significant enrichment. While these enrichments may suggest that the presence of the YFP tag in the lines affects protein synthesis, slowing it to allow the fluorescent protein to fold, we note that in mammalian systems GSK-3β has been shown to complex with chaperones during maturation [48] and to colocalise with chaperone complexes in a Huntington's disease model [49]. Thus, whether these represent noise or biologically relevant interactions remains to be determined, however, we note that no

**Table 1. Significant interacting proteins identified by IPAC-MS for Sgg-PA, Sgg-PB and the Sgg^CPTI115553 protein trap line.**

| Sgg-PA | | | Sgg-PB | | | Sgg-CPTI | | |
|---|---|---|---|---|---|---|---|---|
| **FlyBase ID** | **Protein** | **FDR** | **FlyBase ID** | **Protein** | **FDR** | **FlyBase ID** | **Protein** | **FDR** |
| FBgn0035853 | Ubiquitin conjugating enzyme E2M | 1.42E-03 | FBgn0035621 | uncharacterized protein | 6.50E-05 | FBgn0000181 | Bicaudal | 2.00E-06 |
| FBgn0035621 | uncharacterized protein | 1.45E-03 | FBgn0004649 | Yolkless | 2.44E-04 | FBgn0010774 | RNA and export factor binding protein 1 | 5.48E-03 |
| FBgn0033879 | uncharacterized protein | 3.67E-03 | FBgn0250848 | 26-29kD-proteinase | 3.11E-04 | FBgn0035621 | uncharacterized protein | 2.29E-02 |
| FBgn0029969 | uncharacterized protein | 9.08E-03 | FBgn0285925 | Fasciclin 1 | 3.53E-04 | FBgn0003391 | Shotgun | 2.84E-02 |
| FBgn0029785 | Ribosomal protein L35 | 1.07E-02 | FBgn0285954 | Cabeza | 8.97E-04 | FBgn0010215 | Alpha Catenin | 2.94E-02 |
| FBgn0039857 | Ribosomal protein L6 | 1.10E-02 | FBgn0035909 | Ergic53 | 1.32E-03 | FBgn0030136 | Ribosomal protein S28b | 2.99E-02 |
| FBgn0020660 | eukaryotic translation initiation factor 4B | 1.37E-02 | FBgn0028688 | Regulatory particle non-ATPase 7 | 1.93E-03 | FBgn0015778 | Rasputin | 3.14E-02 |
| FBgn0017579 | Ribosomal protein L14 | 1.44E-02 | FBgn0000181 | Bicaudal | 1.95E-03 | FBgn0041775 | Trailer hitch | 3.16E-02 |
| FBgn0000181 | Bicaudal | 1.62E-02 | FBgn0035208 | uncharacterized protein | 4.15E-03 | FBgn0262743 | Female sterile (2) Ketel | 3.30E-02 |
| FBgn0000308 | Chickadee | 2.19E-02 | FBgn0261397 | dilute class unconventional myosin | 7.41E-03 | FBgn0042134 | Caprin | 3.31E-02 |
| FBgn0003279 | Ribosomal protein L4 | 2.56E-02 | FBgn0003371 | Shaggy | 9.11E-03 | **FBgn0000117** | **Armadillo** | 3.31E-02 |
| FBgn0030086 | Chaperonin containing TCP1 subunit 2 | 2.79E-02 | FBgn0013770 | Cysteine proteinase-1 | 1.19E-02 | FBgn0259173 | Cornetto | 3.38E-02 |
| FBgn0266446 | uncharacterized protein | 2.83E-02 | FBgn0001122 | G protein alpha o subunit | 1.61E-02 | FBgn0003371 | Shaggy | 3.46E-02 |
| FBgn0020235 | ATP synthase, gamma subunit | 2.98E-02 | FBgn0029969 | uncharacterized protein | 1.64E-02 | FBgn0020279 | lingerer | 3.53E-02 |
| FBgn0260639 | Gamma-Tubulin at 23C | 3.34E-02 | **FBgn0000117** | **Armadillo** | 2.64E-02 | FBgn0051716 | CCR4-NOT transcription complex subunit 4 | 3.56E-02 |
| FBgn0024733 | Ribosomal protein L10 | 3.42E-02 | FBgn0010516 | Walrus | 2.77E-02 | FBgn0266557 | Kismet | 3.65E-02 |
| FBgn0011640 | Lark | 3.45E-02 | FBgn0027108 | Innexin 2 | 3.31E-02 | FBgn0034181 | uncharacterized protein | 3.82E-02 |
| FBgn0039713 | Ribosomal protein S8 | 3.90E-02 | FBgn0039704 | Neyo | 3.33E-02 | FBgn0005771 | No ocelli | 4.54E-02 |
| FBgn0001092 | Glyceraldehyde 3 phosphate dehydrogenase 2 | 3.92E-02 | FBgn0032773 | Fondue | 3.61E-02 | FBgn0283479 | Alkaline phosphatase 1 | 4.79E-02 |
| FBgn0010078 | Ribosomal protein L23 | 3.96E-02 | FBgn0030699 | uncharacterized protein | 3.70E-02 | FBgn0030699 | uncharacterized protein | 4.85E-02 |
| FBgn0038805 | Mitochondrial transcription factor A | 4.13E-02 | FBgn0038914 | Female-specific independent of transformer | 4.12E-02 | | | |
| FBgn0260442 | Rhea | 4.18E-02 | | | | | | |
| FBgn0032444 | Chaperonin containing TCP1 subunit 4 | 4.20E-02 | | | | | | |
| FBgn0034968 | Ribosomal protein L12 | 4.50E-02 | | | | | | |
| FBgn0028697 | Ribosomal protein L15 | 4.51E-02 | | | | | | |

(*Continued*)

**Table 1.** (Continued)

| Sgg-PA | | | Sgg-PB | | | Sgg-CPTI | | |
|---|---|---|---|---|---|---|---|---|
| **FlyBase ID** | **Protein** | **FDR** | **FlyBase ID** | **Protein** | **FDR** | **FlyBase ID** | **Protein** | **FDR** |
| FBgn0034654 | Eukaryotic translation initiation factor 3 subunit k | 4.87E-02 | | | | | | |

FlyBase ID represents the gene ID for the corresponding protein. FDR is the false discovery rate calculated by QProt. Red text indicates proteins common to all three pull downs, bold indicates the known target Armadillo.

such enrichments were observed with the Sgg-PB proteoform and loss of Sgg-PA proteoforms does not lead to upregulation of the stress response in our RNA-seq analysis, suggesting there is no general translational disruption in the Sgg-PA and CPTI lines.

In common to the 3 different tagged lines we detected significant enrichment of Bicaudal and CG10591 proteins (red in Table 1). Along with its characterised maternal role in early segmentation, Bicaudal is widely expressed during embryogenesis where it has a role in translation via binding nascent peptides. CG10591 is a protein of unknown function. Encouragingly, with Sgg-PB and the CPTI lines we identified Armadillo (β-Catenin), a known Sgg substrate, as an interacting partner but not with Sgg-PA. With the CPTI line we also identified Shotgun, an E-Cadherin known to bind β-Catenin, and α-Catenin, which also interacts with Arm. Finally, the G protein α o subunit, known to be involved in Wnt signalling, [50] was identified with Sgg-PB. Thus for two of the lines we find evidence for predicted interactions. Specific to the Sgg-PB line we identified the gap junction protein Innexin 2 [51], known to localise with Sgg and E-Cadherin, along with the cell adhesion molecule Fasciclin 1 and Neyo, a component of the zona pellucida complex [52]. We also detected the interaction with Regulatory particle non-ATPase 7, which is involved in the ATP-dependent degradation of ubiquitinated proteins including CACT, an important component for the degradation of NF-kappa-B inhibitor or degradation of Cl that participates in the Hedgehog (Hh) signalling pathway. Together these interactions are consistent with the enrichment of Sgg-PB isoform at the cell membrane and also with known roles for Sgg in regulating aspects of cell junctions. In the case of Sgg-PA, we identified the Talin protein Rhea, the actin binding profilin Chickadee and Gamma Tubulin 23C, cytoskeletal components known to be expressed in the nervous system. Among other interactors of Sgg-PA associated with nervous system localisation we detected Lark [53], which mediates aspects of circadian clock output [54] and two subunits of the Chaperonin TCP complex (2 and 4) that are known to have roles in the nervous system. Together, these identified interacting proteins are consistent with the localisation of Sgg-PA to the CNS. Interestingly, the RNA-seq analysis of $sgg^{isoA}$ mutants identified several misregulated components of the mitochondrial ATP synthesis pathway and our IPAC analysis identified the gamma subunit of ATP synthase along with the mitochondrial regulator TFAM, indicating a potential link with neural energy homeostasis. Taken together, the iPAC analysis identified several known Sgg interacting proteins, found little overlap between proteins enriched with the proteoform-specific pulldowns and indicates that Sgg is involved in diverse, tissue-specific processes in the embryo.

## Lifespan and locomotor dysfunction in $sgg^{isoA}$ mutants

While GSK-3 is a recognized target for the treatment of age related pathologies and multiple diseases, its role in the aging process remains unclear [55]. According to some studies, RNAi knockdown of *sgg* in *Drosophila* shortens lifespan or causes lethality [56]. However, these results are contradictory to expectations from earlier studies suggesting that lithium treatment

extends lifespan via GSK-3 inhibition, determined using specific RNAi to mediate reduction in *sgg* expression [57]. In knock-out mice, loss of GSK-3β is embryonic lethal, whereas GSK-3α null mice exhibit shortened lifespan and increased age-related pathologies [58]. Given the apparent contradictory evidence of the role of *sgg* in lifespan, we investigated whether loss of the Sgg-PA proteoform, which is viable and fertile, positively or negatively influences life span in *Drosophila*.

We performed a standard survival analysis of *sgg*<sup>isoA</sup> null males and females separately, along with matched flies from the progenitor strain. We found that homozygous *sgg*<sup>isoA</sup> females showed significantly reduced survival (~15%, p <0.05) compared to controls and an even more sever reduction (~25%, p<0.05) in hemizygous males (Fig 5A). These results indicate that in flies as in mice, deletion of the Sgg-PA proteoform has a negative effect on longevity.

Since Sgg-PA is extensively expressed in the developing nervous system and GSK-3α mutant mice have nervous system phenotypes, we sought to determine whether the loss of this proteoform has effects on neural function in flies by looking at impairment in locomotor activity via climbing assays (Fig 5B). In homozygous females the loss of Sgg-PA resulted in a 40% decrease in locomotor activity across the lifespan with a similar reduction observed in hemizygous males. The maximum climbing activity was observed in 10 days old flies and was just over 75% for the control line and approximately 45% for the *sgg*<sup>isoA</sup> null. The climbing activity gradually decreased over time and in 40 days old flies was reduced to 55% for the control line and 27% for *sgg*<sup>isoA</sup> null flies (Fig 5B). These observations indicate that loss of the predominantly nervous system expressed Sgg-PA proteoform impairs motor function.

Taken together, our results indicate that although conditional modulation of GSK-3 levels may prolong lifespan or can mitigate the negative age-associated symptoms observed with diseases such as Alzheimer's or diabetes [55], the isoform specific knockout of a nervous system-specific proteoform results in reduced lifespan and locomotor defects. Given the positive impact of GSK-3 inhibition on multiple diseases ranging from neurological disorders to cancer, as well widespread therapeutic interventions targeting GSK-3, further studies are required to assess long-term effects on the aging process and the risks associated with nervous system impairment.

Taken together, our studies indicate that GSK3 performs complex functions mediated by multiple different spliced isoforms that generate functionally distinct proteins. At the level of proteoform expression we provide evidence of complex temporal and tissue specific protein localisation that presumably results from highly regulated tissue-specific splicing as well as evidence for the use of non-canonical transcriptional initiation. Our most striking finding is the major functional differences observed between the two most abundant 3' coding exons, with the short form mediating the well established essential roles for Sgg in development, whereas the long form has nervous specific expression and measurable functional roles in nervous system function. This situation is reminiscent of the divergent roles apparently played by GSK-3α and GSK-3β in mammalian systems. The identification of a different repertoire of interacting proteins for the major *Drosophila* GSK3 proteoforms may provide clues as to the differing roles played by the vertebrate orthologues and opens a route to understanding how this critical kinase can be deployed in different biological contexts.

## Materials and methods

### Cloning gRNAs

To generate the transgenic fly lines carrying the tagged isoforms, we used CRISPR/Cas9 technology as previously described [38]. We initially designed the insertion sites as indicated (Fig

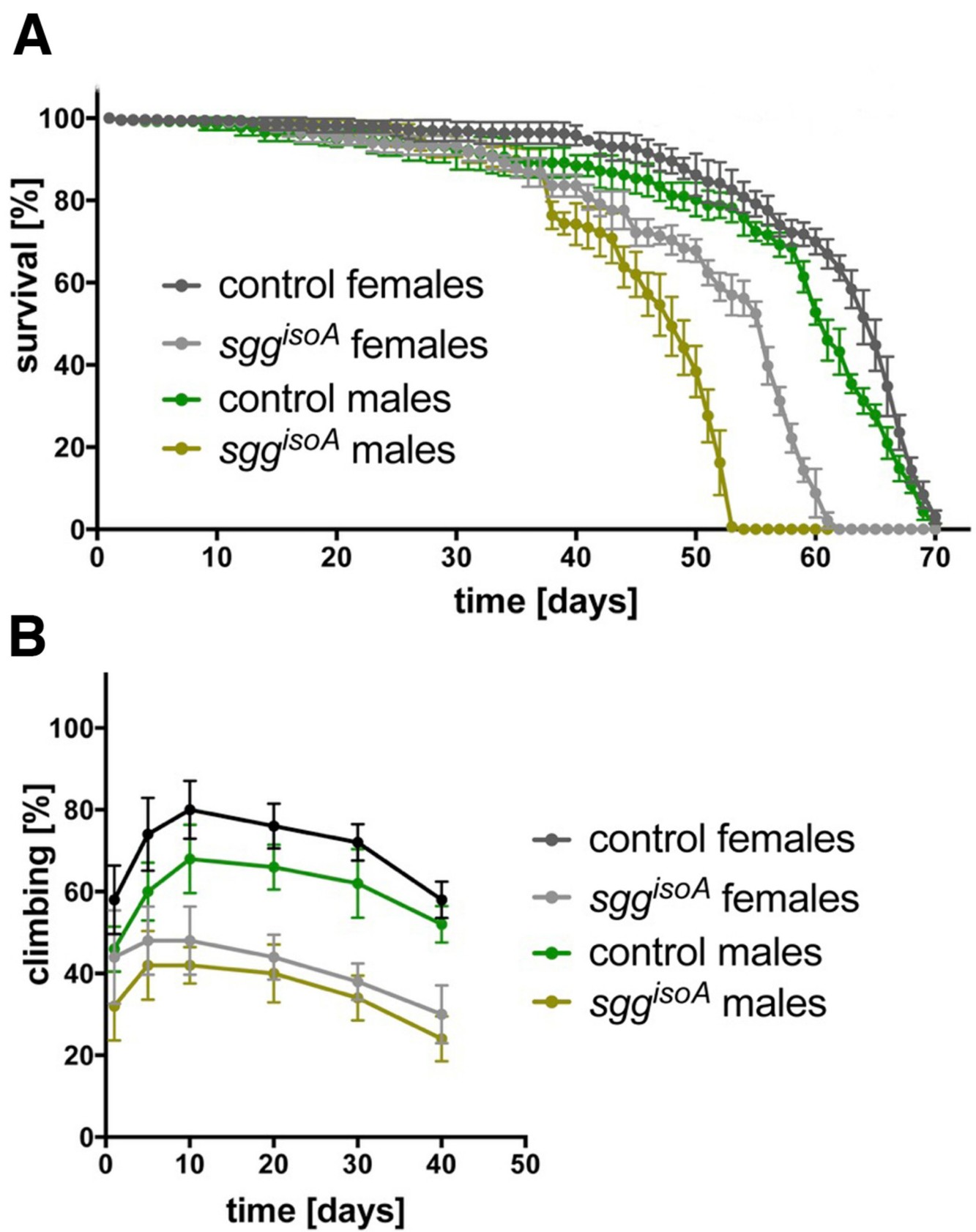

**Fig 5. *sgg^isoA* phenotypic assays. (A)** Graph of lifespan from replicate lines of progenitor controls, *sgg^isoA* null females and males as indicated. Error bars represent standard deviation. **(B)** Graph of locomotor activity as measured by climbing assays with replicate lines of progenitor controls, *sgg^isoA* null females and males as indicated. Error bars represent standard deviation.

1) and choose appropriate gRNAs (S3 Table) for cloning into pCDF3 or pCDF4 vectors [59]. Briefly, target specific sequences were synthesized and either 5′-phosphorylated annealed, and ligated into the *Bbs*I sites of pCDF3 or amplified by PCR for cloning into pCDF4 pre-cut with *Bbs*I.

## Generation of donor vectors

Unless otherwise noted, cloning of donor vectors was performed with the Gibson Assembly Master Mix (New England Biolabs). PCR products were produced with the Q5 High-Fidelity 2X Master Mix (New England Biolabs). All inserts were verified by sequencing. Primers used for plasmid construction are listed in S4 Table.

## Drosophila methods

Embryos were injected using standard procedures into the *THattP40* line expressing *nos*-Cas9 [60, 61]. 500 ng/μl of donor DNA in sterile dH$_2$O was injected together with 100 ng/μl of gRNA plasmid. Individually selected surviving adults were crossed to *w^1118* and the progeny screened for fluorescence: positive flies were balanced and homozygous stocks established where possible (Primers used for genotyping are listed in S5 Table). Injections were performed by the Department of Genetics Fly Facility (https://www.flyfacility.gen.cam.ac.uk). All fly stocks were maintained at 25°C on standard cornmeal medium. Embryos were collected from small cages on yeasted grape juice agar plates.

## Immunostaining

Localization of tagged proteoforms in embryos was visualized by immunohistochemistry using Mouse Anti-FLAG M2 (F1804 Sigma), followed by biotinylated goat anti-Mouse IgG (*BA-9200*, Vector Laboratories) and the Vectastain ABC HRP Kit (PK-4000, Vector Laboratories) using standard protocols [62]. Embryos were mounted in glycerol and imaged using a Zeiss Axiphot.

## Confocal microscopy

For fluorescence imaging, embryos were collected, dechorionated and quickly fixed to avoid bleaching then mounted in glycerol. For live imaging, embryos were dechorionated and mounted in halocarbon oil. Images were acquired using a Leica SP8 confocal microscope (Leica microsystems) with appropriate spectral windows for mVenus and mCherry. images were processed with the Fiji software [63].

## RNA-seq

10-15hr embryos from a homozygous *sgg^isoA* stock and a control line were collected and processed for RNA-seq as described below. In the case of *sgg^isoB*, non-fluorescent embryos from a cross between *sgg^isoB/FM7-GFP* X *FM7-GFP/Y* were collected. In parallel, non-fluorescent embryos from a +/*FM7-GFP* X *FM7-GFP/Y* cross were also collected (FM7-GFP: FM7c, P{GAL4-twi.G}108.4, P{UAS-2xEGFP}AX).

Tissue was homogenised in 300 μl TRIzol with a motorised pellet pestle for 30 seconds. The sample volume was then increased to 1 ml, then 200 μl of Chloroform was added and vortexed.

Samples were centrifuged at max speed for 15 minutes at room temperature and the upper phase transferred to a new 1.5 ml tube. The RNA was then precipitated with 0.8 volumes of Isopropanol and incubated at -20˚C for 2 hours. The samples were then centrifuged at 4˚C at maximum speed for 30 minutes to pellet the RNA. The pellet was then washed with 1 ml 70% ethanol and centrifuged cold for another 5 minutes. After complete removal of the ethanol the RNA was dried for 5 minutes and re-suspended in 20 μl DEPC-treated water. The concentration of samples was determined with the Qubit RNA HS Assay Kit and sample quality assessed with the Bioanalyzer and on 1% agarose gels. Sequencing libraries were prepared using the NEBNext® Ultra™ II Directional RNA Library Prep Kit for Illumina. For each sample 100 ng total RNA were processed with the NEBNext Poly(A) mRNA Magnetic Isolation Module. For all reactions, half volumes were used throughout the protocol, with the exception of the AMPure bead clean-up step where washes were performed used the standard 200 μl 80% fresh Ethanol. Samples were barcoded and PCR amplifications were performed for 12 cycles. After Bioanalyzer quality control equal amount of sample libraries were pooled and submitted for Illumina single-end sequencing.

Fastq reads were aligned using tophat (v2.1.1) with Bowtie (version: 2.3.4.0) using the default parameters. Gene counts tables across samples were created with Rsubread (1.22.3) using dmel_r6.20.gtf and options GTF.featureType = "exon" and GTF.attrType = "gene_id" and default parameters. Read counts per experiment ($sgg^{isoA}$ and $sgg^{isoB}$ experiments were processed independently) were imported into edgeR (3.14.0) and filtered using the filterByExpr function with the default parameters (10397 genes were retained for $sgg^{isoA}$ and 10693 genes for $sgg^{isoB}$). The data was then normalised in limma (3.28.21) using limma-voom. Significant genes were identified fitting a linear model (lmFit) and empirical Bayes method (eBayes) [64–67]. Genes were considered significant differential expressed with fdr $< = 0.05$ and logFC $> = |0.7|$. RNA-seq data are available from GEO (GSE139040).

## iPAC-MS

Embryos from 8–20 hr collections were washed from agar plates with tap water, collected in 100 μm sieves, rinsed in the same solution to remove any yeast, dechorionated in 50% bleach for 1 min, rinsed again and placed on ice. Where necessary, washed embryos were frozen at −80˚C until a sufficient quantity was collected. For each purification, ~200 μl wet-volume of embryos were manually homogenized with a 2 ml Dounce homogenizer in 1 ml of extraction buffer (50 mM Tris, pH 7.5, 125 mM NaCl, 1.5 mM $MgCl_2$, 1 mM EDTA, 5% Glycerol, 0.4% Igepal CA-630, 0.5% digitonin and 0.1% Tween 20) and processed essentially as previously described [47]. Samples were independently immunopurified using StrepII, FLAG and YFP.

ANTI-FLAG® M2 affinity gel (Sigma) and *Strep*-Tactin® Superflow® resin (IBA) were used to capture each FLAG-tagged or StrepII-tagged bait and its binding partners, respectively [47]. For pulldown of fluorescently tagged proteins (YFP), anti-GFP mAb agarose resin (MBL International) was used. Briefly, protein concentration estimation in the embryo lysate was performed using a DC assay (Bio-Rad). The lysate was divided equally into three parts (6 mg total protein per pulldown), to which each resin, pre-washed in extraction buffer, was added. Following 2 h of incubation at 4˚C on a rotating wheel, the resin was washed three times in extraction buffer. Immunoprecipitates were eluted twice each, using 100 μg/ml 3xFLAG peptide (Sigma) in lysis buffer for FLAG immunoprecipitates and 10 mM desthiobiotin in lysis buffer for *Strep*-Tactin immunoprecipitates; each for 10 minutes at 4˚C. Anti-GFP resin immunoprecipitates were eluted in 100 mM glycine-HCl, pH 2.5 with gentle agitation for 30 seconds, followed by immediate neutralization in 1 M Tris-HCl, pH 10.4.

Purification of the baits was confirmed via immunoblots (data not shown) and samples were prepared for mass spectrometric analysis using in-gel digestion, allowing the sample to

enter 2 cm into an SDS-PAGE gel. Gels were fixed and stained with colloidal Coomassie stain, after which the protein-containing band was excised and cut into two equally sized parts. Each band was destained, reduced with dithiothreitol, alkylated with iodoacetamide and subjected to tryptic digest for 16 hours at 37˚C. Approximately 1 μg of peptides from each digested band was analysed using LC-MS/MS on a Q Exactive mass spectrometer (ThermoFisher Scientific), as previously described [68].

For label-free quantification (LFQ), data were processed using MaxQuant (version 1.6.3.4) [69]. Raw data were searched against protein sequences contained in the all translation database obtained from FlyBase release FB2017_06 dmel_r6.19 at (ftp://ftp.flybase.net/releases/FB2017_06/dmel_r6.19/fasta/). The database was customised with the addition of the Uniprot proteome for *Wolbachia pipientis wMel* (https://www.uniprot.org/uniprot/?query=proteome:UP000008215). Within MaxQuant, searches were performed against a reversed decoy dataset and a set of known contaminants. Default search parameters were used with digestion using trypsin allowing two missed cleavages and minimum peptide lengths of six. Carbamidomethyl Cysteine was specified as a fixed modification. Oxidation of Methionine, N-term protein Acetylation and Phosphorylation of Serine, Threonine and Tyrosine were specified as variable modifications. Additionally, "match between runs" was enabled with fractions specified to limit matching to occur only between replicates of the same bait and tag combination.

Identification of interacting partners was performed with QProt [70]. Input for QProt was created from the MaxQuant "proteinGroups.txt" output file. Individually for each bait and tag combination, proteins that were not reverse decoys or potential contaminants were extracted where there was an LFQ reported for at least one tagged replicate. The LFQ values for proteins detected in the tagged bait pulldowns were matched with the corresponding tag in the wild type pulldown. Enrichment analysis was then performed for each bait and tag combination against the corresponding wild type using qprot-param with burn-in set to 10,000 and number of iterations set to 100,000. The QProt tool getfdr was used to calculate the FDR of enrichment. Any proteins enriched in pulldowns using at least two of the three tags with an FDR of less than 0.05 were classed as enriched and the highest FDR from the replicates is reported.

### Lifespan determination

Adult female and male flies were collected shortly after eclosion and separated into 5 cohorts of 100 flies (500 total) for each genotype. Flies were maintained at 25˚C and transferred to fresh food every 2 days at which time the number of surviving flies was recorded.

### Locomotor behaviour

Adult female and male flies were collected shortly after eclosion and separated into 10 cohorts consisting of 10 flies (100 total) for each genotype. Flies were maintained at 25˚C and transferred to fresh food every 3 days. For the climbing assay, each cohort was transferred to an empty glass cylinder (diameter, 2.5 cm; height, 20 cm), and allowed to acclimatize for 5 min. For each trial, flies were tapped down to the bottom of the vial, and the percentage of flies able to cross an 8-cm mark successfully within 10 s was recorded as the climbing index. Five trials were performed for each cohort, with a 1-min recovery period between each trial. Climbing assays were performed 1, 5, 10, 30 and 40 days after eclosion.

### Supporting information

**S1 Fig. (A-C)** anti-FLAG immunohistochemistry of Sgg-PA expression in the ventral neuroectoderm of a stage 10 embryo **(A)**, the brain from a stage 16 embryo **(B)** and the head region showing various sensory organs **(C)**. Scale = 20μm. **(D and E)** dorsal view of a stage 16

Sgg-PB embryo showing expression in the gut (arrows) and mesoderm (arrowheads), Scale = 100μm, and lateral view of hindgut **(E)** Scale = 20μm. **(F)** blastoderm embryo expressing mCherry tagged Sgg-PB and YFP tagged Sgg-PA, demonstrating ubiquitous Sgg-PB, the YFP signal represents yolk cell autofluorescence, Scale = 100μm. **(G)** close up of punctate epidermal expression in a Sgg-PB mCherry stage 16 embryo, Scale = 20μm.
(PDF)

**S2 Fig. Immunohistochemistry of tagged Sgg proteoforms. A-U)** Anti-flag staining of embryos with the indicated tagged proteoforms at stage 5–6 (blastoderm, left column), 10–11 (germband extension, middle column) and 16 (late embryogenesis, right column). All embryos oriented anterior to the left dorsal to the top except U, which is a dorsal view. See text for full details of expression. **(D and E)**, arrows = meseoderm; **(F)** arrow = salivary gland, arrowheads = malphigian tubules, asterisk = posterior spiracles; **(H)** arrow = mesoderm; **(I)** arrow = hindgut; **(L)** arrow = hindgut, arrowhead = foregut, white arrowhead = pharynx; **(N)** arrow = mesoderm; **(O)** arrow = hindgut, arrowhead = proventriculus; **(Q)** arrow = mesoderm; **(R)** arrow = hindgut, arrowhead = anterior midgut; **(U)** arrow = salivary gland, arrowhead = proventriculus. Scale bar in A = 100μm. **(V-Y)** Close up dorsal views highlighting: **(V)** Sgg-PD expression in the salivary gland (arrow) and proventriculus (arrowhead); **(W)** Sgg-PG in the foregut (arrowhead) and anterior region of the pharynx (arrow); **(X)** prominent Sgg-PO expression in the proventriculus (arrow) and **(Y)** Sgg-PP in the hindgut (arrow). Scale bar in V = 20μm applies to V-Y.
(PDF)

**S1 Table. Protein and transcript isoforms encoded by the *D. melanogaster sgg* locus.**
(TXT)

**S2 Table. Transcripts with significant expression changes in *sgg*^isoA and *sgg*^isoB embryos compared with stage matched progenitors.**
(TXT)

**S3 Table. List of gRNAs used to generate CRISPR/Cas9 mediated HDR and transgenic fly lines.**
(XLSX)

**S4 Table. List of primers used to generate donor vectors for homology mediated recombination via CRISPR/Cas9.**
(TXT)

**S5 Table. List of primers used for genotyping engineered flies.**
(TXT)

## Author Contributions

**Conceptualization:** Simon Hubbard, Kathryn Lilley, Steven Russell.

**Data curation:** Dagmara Korona, Daniel Nightingale, Bertrand Fabre, Michael Nelson, Bettina Fischer.

**Formal analysis:** Dagmara Korona, Daniel Nightingale, Bertrand Fabre, Bettina Fischer, Jonathan Lees, Steven Russell.

**Funding acquisition:** Simon Hubbard, Kathryn Lilley, Steven Russell.

**Investigation:** Dagmara Korona, Daniel Nightingale, Michael Nelson, Bettina Fischer, Glynnis Johnson, Steven Russell.

**Methodology:** Dagmara Korona, Daniel Nightingale, Bertrand Fabre, Glynnis Johnson, Jonathan Lees.

**Project administration:** Simon Hubbard, Kathryn Lilley, Steven Russell.

**Software:** Michael Nelson.

**Supervision:** Simon Hubbard, Kathryn Lilley, Steven Russell.

**Validation:** Dagmara Korona, Daniel Nightingale, Steven Russell.

**Writing – original draft:** Dagmara Korona, Steven Russell.

**Writing – review & editing:** Dagmara Korona, Daniel Nightingale, Bertrand Fabre, Michael Nelson, Bettina Fischer, Glynnis Johnson, Jonathan Lees, Simon Hubbard, Kathryn Lilley.

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
