## [Decision Letter · Decision Letter 0]

24 Jun 2020

PONE-D-20-12754

Characterisation of protein isoforms encoded by the Drosophila Glycogen Synthase Kinase 3 gene shaggy

PLOS ONE

Dear Dr. Russell,

Thank you for submitting your manuscript to PLOS ONE. After careful consideration, we feel that it has merit but does not fully meet PLOS ONE’s publication criteria as it currently stands. Therefore, we invite you to submit a revised version of the manuscript that addresses the points raised during the review process.

The two reviewers have made a number of comments that should be addressed and that I think will greatly enhance the impact of what is likely to be an important contribution.  Please address them in your revision - particularly with regards that quality and interpretability pointed out by review #1.   

We look forward to receiving your revised manuscript.

Kind regards,

Michael Klymkowsky, Ph.D.

Academic Editor

PLOS ONE

Journal Requirements:

3. Please include a copy of Table 1 which you refer to in your text on page 8 and 9.

Reviewers' comments:

Reviewer's Responses to Questions

**Comments to the Author**

1. Is the manuscript technically sound, and do the data support the conclusions?

Reviewer #1: No

Reviewer #2: Yes

2. Has the statistical analysis been performed appropriately and rigorously? 

Reviewer #1: I Don't Know

Reviewer #2: Yes

3. Have the authors made all data underlying the findings in their manuscript fully available?

Reviewer #1: No

Reviewer #2: Yes

4. Is the manuscript presented in an intelligible fashion and written in standard English?

Reviewer #1: Yes

Reviewer #2: Yes

5. Review Comments to the Author

Reviewer #1: This manuscript bravely addresses the complexity of the shaggy (Sgg) gene, which encodes multiple variations of the Drosophila GSK-3 primarily through alternative splicing. The authors have knocked in tagged versions for each splice form, examined protein expression patterns for multiple splice forms, and compared phenotypes for loss of the two major proteoforms. This work may provide insights into the functions of GSK-3 across species and may help to define the roles of Sgg in development, lifespan, and motor function. However, the manuscript could be revised to improve clarity, improve documentation of the figures and data, and provide standard controls that they likely have in hand already.

1. Major concerns are the quality of the IHC images in Figures 2 and 4. Figure 2 has no controls for background staining and it is difficult to discern signal from background in many of the panels. The figure needs a negative control such as no primary antibody, especially for Fig2J-O, reportedly showing Sgg-PB expression. For example, Fig 2, Panel J (presumably ubiquitous expression) is difficult to distinguish from Fig 2, panel A (presumably background). The figures are also not well annotated; arrows or other annotation of the figures would helpful for these less than obvious expression patterns. The magnification changes from one panel to the next, so they should also include scale bars for every photomicrograph.

2. Figures 2G and 2H reportedly show cytoplasmic expression that was “particularly concentrated in the vicinity of the cell membrane.” This subcellular pattern is not clear. It is not even clear (to this observer) that staining in panel 2G is intracellular vs trapped between what appear to be a cluster of cells. Protein localization data should be improved with higher resolution methods. For example, a higher resolution confocal image with a counter stain for cellular landmarks (plasma membrane and nuclear markers for example) would be helpful to make their conclusions more convincing.

3. This is a stylistic suggestion, not a critique: Why not rearrange figures 2 and 3 to pair Flag/IHC (Fig 2) with YFP/IF (Fig 3 A-D) by proteoform rather than sorting by imaging method? For Figure 3E-I, they could then show the YFP and mCherry as separate images as well as showing the overlap. This is just a suggestion that I would leave up to the authors.

4. The authors should discuss the original work of Ruel et al (Nature 1993) showing that mammalian Gsk3b can rescue sgg null phenotypes, which seems to suggest that alternatively spliced forms are not essential. Perhaps this is a more nuanced point but this point should be discussed (Ruel et al was cited but this point was not discussed).

5. Figure 4, as in Figure 2, does not have a negative control, and it is difficult to distinguish background from real staining. Furthermore, relevant structures are not highlighted, there are no scale bars, and more generally, this is not a particularly informative or useful figure. They could remove this figure without detriment to the manuscript and just describe the results in the text, or, if the journal allows, move this to supplementary data (but would still need background control, scale bars, and clearer labeling of structures).

6. Table 1 with a list of interacting proteins was not in the downloaded PDF. This review cannot be completed until all data are available.

Minor comments:

7. Gene names in Figure 5B are pixelated and illegible.

8. Rather than referring to developmental stages by numbers, the authors could use more descriptive names for the stages shown so that readers unfamiliar with Drosophila staging can appreciate the work more readily?

9. Page 3: “in Xenopus, knockdown results in axis formation defects [11].” These experiments involved expression of a kinase-dead mutant GSK-3; they were not knockdowns. Similar “dominant-negative” experiments in Xenopus were reported by He et al and by Pierce et al.

10. “In contrast to vertebrates, the Drosophila genome contains a single GSK-3 locus…” What about Gasket, similar to GSK-3 and sgg but expressed from a distinct locus?

Reviewer #2: This study follows up some 30 years after the original publications of the cloning of shaggy/zeste-white3 by the Simpson and Perrimon groups and uses insertions of tags into various exons to trace the various potential isoforms of the single shaggy gene. The authors then describe the expression patterns of the protein variants, demonstrating that they exist and also perform some proteomic analysis. They also selectively delete the two major isoforms, Sgg-PA and Sgg-PB to evaluate the effect on viability, longevity and in a behavioural test. This is a complete and compelling analysis that significantly enhances understanding of the shaggy locus.

Minor points:

1. The anti-FLAG immunohistochemistry in figure 2 has a significant background and it would be advantageous to show immunoblotting of the isoforms from the embryos to indicate the relative levels of expression of the different proteoforms. This would also be useful to do for the Sgg-PA and Sgg-PB knockouts (se evaluate any interactive effect on expression).

2. The authors perhaps go too far in comparing the Sgg-PA and Sgg-PB proteoforms with the two mammalian isoforms encoded by different genes). In the latter case, both are ubiquitously expressed and the similarities may be coincidental.

6. PLOS authors have the option to publish the peer review history of their article (what does this mean?). If published, this will include your full peer review and any attached files.

Reviewer #1: No

Reviewer #2: Yes: Jim Woodgett

---

## [Author Response · Author response to Decision Letter 0]

8 Jul 2020

We thank the reviewers for their time and effort on our manuscript and the helpful comments. Below we address the points raised by the editorial team and each reviewer. We hope that the revisions to the manuscript and figures along with the response below are satisfactory.

Kind regards

Steve Russell (on behalf of the authors) 

Journal

1. Formatted as requested

2. “Data not shown” removed, the statement is not necessary since the very phenotypes alluded to are minor and we regard them as insignificant. We have deposited the images on out University data store https://doi.org/10.17863/CAM.54850 .

3. Table 1 included, apologies for the omission.

1. Major concerns are the quality of the IHC images in Figures 2 and 4. Figure 2 has no controls for background staining and it is difficult to discern signal from background in many of the panels. The figure needs a negative control such as no primary antibody, especially for Fig2J-O, reportedly showing Sgg-PB expression. For example, Fig 2, Panel J (presumably ubiquitous expression) is difficult to distinguish from Fig 2, panel A (presumably background). The figures are also not well annotated; arrows or other annotation of the figures would helpful for these less than obvious expression patterns. The magnification changes from one panel to the next, so they should also include scale bars for every photomicrograph.

We note that 2 A, D and G shows anti-FLAG staining in wild type embryos with no tags, this provides the standard background control for the specificity of the antibody (also see response to reviewer 2 below). The Sgg-PB protein and transcript have a well described strong maternal contribution. In support of the antibody staining we now include a live confocal image of the mCherry tagged Sgg-PB at the blastoderm stage confirming strong and ubiquitous staining. With respect to the epidermal and mesodermal staining with Sgg-PB, we include Sgg-PB[YFP] close up of the mesoderm (Fig 2O) and Sgg-PB[mCh] close up of the epidermis, again supporting results with the FLAG. Finally, in Fig 2J we show a wild type embryo imaged in the YFP channel showing no signal (apart from well known gut autofluorescence). Taken together we believe the combination of anti-FLAG immunohistochemistry and live confocal imaging are consistent with the localisations we report.

We have added annotations and scale bars to all figures

2. Figures 2G and 2H reportedly show cytoplasmic expression that was “particularly concentrated in the vicinity of the cell membrane.” This subcellular pattern is not clear. It is not even clear (to this observer) that staining in panel 2G is intracellular vs trapped between what appear to be a cluster of cells. Protein localization data should be improved with higher resolution methods. For example, a higher resolution confocal image with a counter stain for cellular landmarks (plasma membrane and nuclear markers for example) would be helpful to make their conclusions more convincing.

Given the current limitations we face in the laboratory, we have elected to remove the statement relating to membrane association and on page 5 have changed the text to read “Looking more closely we observed that in the chordotonal organs staining was associated with the cell bodies and extended into the ciliated endings. (arrows in Figs 2M and P).“

3. This is a stylistic suggestion, not a critique: Why not rearrange figures 2 and 3 to pair Flag/IHC (Fig 2) with YFP/IF (Fig 3 A-D) by proteoform rather than sorting by imaging method? For Figure 3E-I, they could then show the YFP and mCherry as separate images as well as showing the overlap. This is just a suggestion that I would leave up to the authors.

We thank the reviewer for these comments and suggestions and have rejigged the figures. Fig 2 now shows the immunohistochemistry and live confocal images of Sgg-PA and Sgg-PB. Scalebars and arrows have been added for clarity. Some more detailed images are provided in Sup Fig 1

The new Fig 3 included the reciprocal labelling and higher magnification views, scale bars have been added.

The old Figure 3 detailing expression of the other tagged proteoforms has been moved to Supp Fig 2 and has been annotated.

4. The authors should discuss the original work of Ruel et al (Nature 1993) showing that mammalian Gsk3b can rescue sgg null phenotypes, which seems to suggest that alternatively spliced forms are not essential. Perhaps this is a more nuanced point but this point should be discussed (Ruel et al was cited but this point was not discussed).

The rescue experiments in the Ruel et al paper are not straightforward, while the assays used show that a Sgg-PB proteoform (Sgg10) can fully rescue the sggD127 null allele and sggb12 (an allele lacking some but not all isoforms), Gsk-3b (but not Gsk-3a) only provides partial rescue.

We modified the text on Pg 4 to insert: “Rescue experiments in Drosophila indicate that expression of a Sgg-PB isoform can fully rescue sgg null phenotypes and that mammalian GSK-3β but not GSK-3α can provide partial rescue of some, but not all null phenotypes, emphasising the difference between the mammalian genes and pointing to functional differences between Sgg proteoforms [27].”

5. Figure 4, as in Figure 2, does not have a negative control, and it is difficult to distinguish background from real staining. Furthermore, relevant structures are not highlighted, there are no scale bars, and more generally, this is not a particularly informative or useful figure. They could remove this figure without detriment to the manuscript and just describe the results in the text, or, if the journal allows, move this to supplementary data (but would still need background control, scale bars, and clearer labeling of structures).

As suggested we have moved this to Supplementary Figure 2, added scale bars and included some annotations. The negative control remains wild type embryos stained with anti-FLAG

6. Table 1 with a list of interacting proteins was not in the downloaded PDF. This review cannot be completed until all data are available.

We profusely apologise for this oversight

Minor comments:

7. Gene names in Figure 5B are pixelated and illegible.

We have generated a much higher resolution version and apologise for the loss of resolution during file conversion.

8. Rather than referring to developmental stages by numbers, the authors could use more descriptive names for the stages shown so that readers unfamiliar with Drosophila staging can appreciate the work more readily?

While we note this is standard nomenclature in Drosophila (much like vertebrate embryologists use number of somites),we added clarifications, particularly in the figure legends.

9. Page 3: “in Xenopus, knockdown results in axis formation defects [11].” These experiments involved expression of a kinase-dead mutant GSK-3; they were not knockdowns. Similar “dominant-negative” experiments in Xenopus were reported by He et al and by Pierce et al.

We changed the text on page 3 to “expression of a kinase dead version of GSK-3β resulted in axis formation defects [11]”

10. “In contrast to vertebrates, the Drosophila genome contains a single GSK-3 locus…” What about Gasket, similar to GSK-3 and sgg but expressed from a distinct locus?

Pg 3 – inserted “A second GSK-3 is encoded by the gasket locus, a retrotransposed gene whose expression is largely restricted to the testis (Kalamegham, R et al 2007), and is not considered further here”

Reviewer #2: This study follows up some 30 years after the original publications of the cloning of shaggy/zeste-white3 by the Simpson and Perrimon groups and uses insertions of tags into various exons to trace the various potential isoforms of the single shaggy gene. The authors then describe the expression patterns of the protein variants, demonstrating that they exist and also perform some proteomic analysis. They also selectively delete the two major isoforms, Sgg-PA and Sgg-PB to evaluate the effect on viability, longevity and in a behavioural test. This is a complete and compelling analysis that significantly enhances understanding of the shaggy locus.

Minor points:

1. The anti-FLAG immunohistochemistry in figure 2 has a significant background and it would be advantageous to show immunoblotting of the isoforms from the embryos to indicate the relative levels of expression of the different proteoforms. This would also be useful to do for the Sgg-PA and Sgg-PB knockouts (se evaluate any interactive effect on expression).

We are unable to complete satisfactory immunoblotting experiments due to current limitations at our institution, we have pilot data, not suitable for publication, that supports our analysis. It shows a cross-reacting band in wild type with anti-FLAG that is not present with StrepII detection, presumably accounting for the weak background we show. These preliminary data are available from the University data repository (https://doi.org/10.17863/CAM.54850). Taken together, we strongly suggest that the fluorescent imaging – which is clearly independent of the antibody staining – is entirely consistent with the localisations we report and that the preliminary Western data further support the validity of the images we wish to publish. 

2. The authors perhaps go too far in comparing the Sgg-PA and Sgg-PB proteoforms with the two mammalian isoforms encoded by different genes). In the latter case, both are ubiquitously expressed and the similarities may be coincidental.

It may be that the similarities are coincidental, however, the evidence we describe indicates that GSK-3α and GSK-3β are not functionally equivalent: GSK-3α mutants are lethal and are obviously not rescued by GSK-3β, and we note that the work reported in Barrell et al (2012) Novel Reporter Alleles of GSK-3α and GSK-3β. (https://doi.org/10.1371/journal.pone.0050422) indicates that the mouse isoforms show differential expression.

The presence of a related amino-acid extension on Sgg-PA and GSK-3a, the non-essential nature of null mutations inthese proteoforms and the nervous system phenotypes, we believe are worth forwarding the hypothesis that they may be related.

---

## [Decision Letter · Decision Letter 1]

13 Jul 2020

Characterisation of protein isoforms encoded by the Drosophila Glycogen Synthase Kinase 3 gene shaggy

PONE-D-20-12754R1

Dear Dr. Russell,

We’re pleased to inform you that your manuscript has been judged scientifically suitable for publication and will be formally accepted for publication once it meets all outstanding technical requirements.

Kind regards,

Michael Klymkowsky, Ph.D.

Academic Editor

PLOS ONE

Additional Editor Comments (optional):

Reviewers' comments:

Reviewer's Responses to Questions

**Comments to the Author**

1. If the authors have adequately addressed your comments raised in a previous round of review and you feel that this manuscript is now acceptable for publication, you may indicate that here to bypass the “Comments to the Author” section, enter your conflict of interest statement in the “Confidential to Editor” section, and submit your "Accept" recommendation.

Reviewer #1: All comments have been addressed

Reviewer #2: All comments have been addressed

2. Is the manuscript technically sound, and do the data support the conclusions?

Reviewer #1: Yes

Reviewer #2: Yes

3. Has the statistical analysis been performed appropriately and rigorously? 

Reviewer #1: I Don't Know

Reviewer #2: Yes

4. Have the authors made all data underlying the findings in their manuscript fully available?

Reviewer #1: Yes

Reviewer #2: Yes

5. Is the manuscript presented in an intelligible fashion and written in standard English?

Reviewer #1: Yes

Reviewer #2: Yes

6. Review Comments to the Author

Reviewer #1: The authors completely addressed the concerns that I had raised. .

Reviewer #2: The authors have addressed or explained the issues raised (e.g. difficulty in performing experiments due to the pandemic).

Minor note; the DOI link for the preliminary data was broken (https://doi.org/10.17863/CAM.54850) and the GSK3 alpha lethality is beta lethality - but this error is only in the response to reviewers letter, not the manuscript, which has it the right way around.

7. PLOS authors have the option to publish the peer review history of their article (what does this mean?). If published, this will include your full peer review and any attached files.

Reviewer #1: No

Reviewer #2: **Yes: **Jim Woodgett

---

## [Editor Report · Acceptance letter]

14 Jul 2020

PONE-D-20-12754R1 

Characterisation of protein isoforms encoded by the Drosophila Glycogen Synthase Kinase 3 gene shaggy 

Dear Dr. Russell:

I'm pleased to inform you that your manuscript has been deemed suitable for publication in PLOS ONE. Congratulations! Your manuscript is now with our production department. 

Kind regards, 

on behalf of

Dr. Michael Klymkowsky 

Academic Editor

PLOS ONE